# ADVERSARIAL ROBUSTNESS OF COUNT-MIN SKETCH

## ABSTRACT

Small–space frequency estimators play a crucial role in a multitude of settings related to both machine learning and data processing for evolving data. Many frequency estimators use internal randomness to compress the information about the frequencies of items to a small sketch that can be used to provide estimates. Historically, these types of estimators were designed without considering the scenario in which the user with access to the estimator can accidentally or maliciously manipulate estimates. This can be achieved by the user who makes adaptive updates and uses queries to gain information about the estimator's internal randomness.

In this work, we consider one of the simplest such estimators: Count-Min Sketch. On the one hand, we show how to make it resistant to adversarial attacks in both the random oracle model, which corresponds to cryptographically hard hash functions, and using universal hash functions if the domain size is in polynomial relationship with with the size of hash tables.

On the other hand, we also explore adaptive attacks on Count-Min Sketch. In particular, we show how to speed up multirow hashing attacks for a popular family of universal hash functions.

## 1 INTRODUCTION

Frequency estimation and heavy hitter reporting are fundamental data processing problems, in which one wants to approximately track frequencies of items in an evolving multiset—with items arriving and departing—in small space. In this setting, explicitly storing counts of all items would require maintaining a prohibitively large data structure, which would not fit in the main memory or would require devoting a large amount of it, since information theoretically this solutions requires space at least proportional to the number of distinct elements in the data set. Instead, techniques used in this type of scenario often utilize randomness, including randomized projections of the high–dimensional frequency vector into a low dimensional sketch, sampling, and randomly partitioning the elements of the domain into buckets. Examples of well known frequency estimation algorithms Count-Min Sketch (Cormode and Muthukrishnan (2005)), Count Sketch (Charikar et al. (2004)), and Multistage–Filter (Estan and Varghese (2002)).

Applications of frequency estimation span various types of machine learning and data analysis, especially in the dynamic environments in which it is important to adapt to evolving data. Specific examples include semi-supervised learning (Talukdar and Cohen (2014)), natural language processing (Goyal et al. (2012)), feature selection (Aghazadeh et al. (2018)), ranking (Dzogang et al. (2015)), network traffic measurement (Estan and Varghese (2002)), PageRank computation (Sarlós et al. (2006)), and clustering (Spiegel and Polyzotis (2006)).

In this work, we focus on Count-Min Sketch (CMS), one of the simplest and most popular frequency estimators. Off–the–shelf implementations of Count-Min Sketch are available in many data processing systems, including Apache Spark, Databricks, and Twitter Algbird.

**Adversarial robustness.** One shortcoming of randomized estimators is that their results can potentially be manipulated by an adaptive adversary who uses information obtained from queries, which may leak information about the algorithm's internal randomness. In this setting, the traditional analysis approach that assumes a fixed sequence of updates and queries does not apply anymore. In fact, it is known that some types of techniques, including linear sketches, can be vulnerable to adaptive attacks (Hardt and Woodruff (2013)).

To address this shortcoming of small–space algorithms, Ben-Eliezer et al. (2022b) introduced a new framework of *adversarially robust data streaming algorithms*. In this model, the computation can be seen as a game between the algorithm and adversary, in which the adversary can make arbitrary updates and queries. The adversary wins if at least one of the estimates provided in reply to a query is not sufficiently good and the algorithm wins if all estimates are sufficiently good.[1] Ben-Eliezer et al. (2022b) introduced very generic techniques that allow for making many streaming algorithms adversarially robust. Unfortunately, their techniques do not yield good bounds for data streams with arbitrary sequences of insertions and deletions. This was to some extent addressed by Hassidim et al. (2022), who introduced differential privacy to the mix of useful techniques. However, in order to make CMS adversarially robust and provide $Q$ estimates overall, their approach requires aggregating estimates from $\Omega(\sqrt{Q})$ independent instances of CMS, which can be a significant overhead in both the amount of required space and processing time.

**Our results: securing CMS.** We call a CMS instance adversarially robust in our setting if with probability $1 - \delta$, the frequency of no element is overestimated by more than $O(\epsilon\|\boldsymbol{a}\|_1)$, where $\boldsymbol{a}$ is the frequency vector of the elements.

We show how to make CMS adversarially robust in two settings. First, assuming that all internal hash functions are fully random and their values are completely unpredictable to the adversary (this is known as the random oracle model), we show that setting the internal parameters of CMS to $O(\log(Q/\delta)/\epsilon)$ hash tables (which we refer to as *rows*) and $O(1/\epsilon)$ buckets per row suffices, assuming that the adversary performs at most $Q$ operations. This result is useful in the setting with cryptographically secure hash functions that are hard to predict by a computationally–limited adversary. This is a significant improvement over the approach of Hassidim et al. (2022), which would introduce an overhead of $\Omega(\sqrt{Q})$, as we are able to achieve these types of guarantees with an overhead of only $O(\log Q)$.

Our second result is unconditional: it does not rely on cryptographic assumptions and applies to any family of universal hash functions, which are widely used in practice. If $S$ is the size of the domain of elements whose frequencies CMS estimates, we show that with probability $1 - \delta$ over the initial selection of hash functions, all queries provided by CMS work for *any* set of items with $O(\epsilon^{-1}k \log_S(1/\delta))$ rows and $O(S^{1/k}/\epsilon)$ buckets per row, where $k$ is an arbitrary positive integer. By setting $k = \Theta(\log S)$, this implies that, roughly, $O(\epsilon^{-1} \log S \cdot \log_S(1/\delta))$ rows and $O(1/\epsilon)$ buckets per row are sufficient to achieve adversarial robustness over an arbitrary sequence of insertions and deletions (and hence, it has no dependency on $Q$, the number of operations). One may prefer using smaller values of $k$, however, because the running time of CMS queries and updates is proportional to the number of rows.

Both these results are obtained via the same approach that relies on the fact that if no two elements (used by the adversary) collide in a large fraction of hash tables, then all estimates have to be close to the real values.

**Our results: attacking CMS.** A CMS instance maintains $R$ rows, which are hash tables with $B$ buckets each and each bucket containing a single counter. We show that for fully random hash functions (and more generally if the outputs of the hash function land in each bucket with a considerable frequency), we can efficiently find a small set of elements that, when inserted into the CMS, increase the frequency estimate of a selected element *target* without ever inserting *target*. In the traditional non-adversarial analysis, the additive error of an estimate is bounded by $O(\|\boldsymbol{a}\|_1/B)$ with probability $1 - 2^{\Omega(R)}$ (where $\boldsymbol{a}$ is the frequency vector at the time of the query) and often, $R$ is selected to be smaller than $B$. Here we show efficient techniques that go beyond the more straightforward approach of Markelon et al. (2023) to find $R$ elements different from *target* to make the CMS instance overestimate *target*'s frequency by close to $\|\boldsymbol{a}\|_1/R$, even for large values of $B$.

Additionally, we develop more sophisticated attacks that involve the popular family of universal hash functions $h(x) = ((ax + b) \bmod P) \bmod B$ for a large prime $P$. We show that it is possible to recover the internal randomness of these hash functions to perform more efficient multirow attacks

---

[1]Note that Ben-Eliezer et al. (2022b) considered a single function of the stream that the algorithm is supposed to track and the adversary only provides updates. Our setting can be modeled as the entire sequence of updates and queries being the stream and the function providing a non-trivial value only if the last element of the stream is a query.

that lead to significantly worse estimates. We run experiments to demonstrate the practicality of our methods in breaking CMS estimates in Apache Spark.

**Related results.** Most closely related to our results are papers by Clayton et al. (2019) and Markelon et al. (2023), in which they consider adversarial attacks on Count-Min Sketch and related randomized data structures. Our attacks on Count-Min Sketch significantly go beyond theirs by taking advantage of function outputs and speeding up multirow collisions that result in significantly worse estimates for a class of very popular universal hash functions. Both Clayton et al. (2019), Markelon et al. (2023), and Kim et al. (2023) propose modifications to make frequency estimators robust, but they generally cannot handle deletions, i.e., operate in the insertion–only model. We discuss the relationship to these works in more detail in Appendix C.

There also exist deterministic frequency estimation algorithms (Misra and Gries (1982); Demaine et al. (2002)) that are trivially adversarially robust, since they use no randomness. They are limited in that they do not allow for deletions of items, which randomized frequency estimators can often handle.

Cohen et al. (2022) and Cohen et al. (2023) explore the adversarial robustness of Count Sketch (Charikar et al. (2004)), which is closely related to CMS. Count Sketch also hashes elements into buckets and independently computes the estimates for them, which it later combines into estimates for point queries. Count Sketch is nevertheless siginificantly different from CMS and their attack techniques differ substantially from ours and do not easily apply to Count-Min Sketch. Unfortunately, our techniques for making CMS adversarially robust are not sufficient for making Count Sketch adversarially robust. Count Sketch crucially depends on the correctness of $\ell_2$ estimators for its buckets and we do not know how to achieve their adversarial robustness without a significant overhead in the amount of space for bucket estimators (see Ben-Eliezer et al. (2022a)).

## 2 REVIEW OF THE FREQUENCY ESTIMATION PROBLEM AND COUNT-MIN SKETCH

**Preliminaries.** We write $[k]$, where $k \in \mathbb{N}$, to denote $\{0, 1, \ldots, k-1\}$, the set of the first $k$ natural numbers.[2] Moreover, all arrays in our pseudocode are zero–based.

**Frequency estimation.** Before we describe Count-Min Sketch and its properties, let us briefly introduce the problem that Count-Min Sketch addresses. Let $\Omega$ be a domain of elements under consideration. The goal is to create a (small–space) data structure for tracking an evolving *multiset $S$* consisting of elements from $\Omega$. Initially, $S$ is empty. Every update can either add an element to $S$ or remove it from it, but it is not allowed to remove an element more times than it has been inserted. The user of the data structure, apart from inserting and deleting elements, is also allowed to make queries about the number of times a given element occurs in $S$. This type of query is called a *point query*.

To be more formal without making our presentation too complicated, we assume throughout the paper that $\Omega = [|\Omega|]$, i.e., that it consists of the first $|\Omega|$ natural numbers. Everything we say about the properties of Count-Min Sketch in this context equally applies to other types of finite sets. Now, in particular, we can represent $S$ as a *frequency vector* $\boldsymbol{a} = (a_0, a_1, \ldots, a_{|\Omega|-1}) \in \mathbb{R}_{\geq 0}^{|\Omega|}$, where each $a_i$ is the current count of elements $i$. This also allows for extending the definition to fractional values. Now we can think of any update as a pair of numbers $(x, \Delta) \in |\Omega| \times \mathbb{R}$ indicating that $\Delta$ should be added to coordinate $x$, $a_x$. At no point, however, is any coordinate allowed to become negative, which means that $\Delta + a_x \geq 0$ for any update before it gets applied. In the streaming literature, this is known as the *strict turnstile* model. Values of $\Delta = 1$ and $\Delta = -1$ correspond to inserting and deleting a single element, respectively. Each point query is an integer $i \in [|\Omega|]$ indicating the coordinate, $a_i$, for which the estimate is to be provided. For Count-Min Sketch, the desired goal is to return an estimate $\widehat{a_i}$ such that for some parameter $\epsilon \in (0, 1)$: $a_i \leq \widehat{a_i} \leq a_i + \epsilon \|\boldsymbol{a}\|_1$. We refer to this guarantee as an $\epsilon$–*estimate*.

---

[2] We diverge here from the more popular one–based version of this notation, in which $[k] = \{1, 2, \ldots, k\}$. The zero–based version makes more sense in the context of the modular arithmetic that we use in this paper.

**The Count-Min Sketch construction.** We now briefly describe how Count-Min Sketch works. We present its full pseudocode (Algorithm 1), as well as proofs of its properties in Appendix A. An instance of Count-Min Sketch takes two parameters $R \in \mathbb{Z}_+$, the number of rows, and $B \in \mathbb{Z}_+$, the number of buckets per row. Each of the $R \times B$ buckets is initially set to 0. Additionally, for each row $r \in [R]$, Count-Min Sketch generates a hash function $h_r : \Omega \to [B]$ that is used to map elements of the domain to buckets in this specific row. When an element $x$ is inserted or deleted, the count of each bucket to which $x$ is mapped, one per row, is increased or decreased by one, respectively. (This easily generalizes to increments or decrements by an arbitrary number, which corresponds to inserting and deleting multiple elements at once.) To provide an estimate $\widehat{a_x}$ in reply to a point query about $x$, one returns the minimum of counts in all buckets to which $x$ is assigned.

**Properties of Count-Min Sketch in the non-adaptive environment.**

**Fact 1.** *The result of a point query in CMS is never an underestimate of the true frequency.*

**Fact 2.** *Let $\epsilon \in (0, 1)$ and $\delta \in (0, 1)$. Consider a CMS instance with $R \geq \lceil \frac{e}{\epsilon} \rceil$ rows and $B \geq \lceil \ln(1/\delta) \rceil$ buckets per row with each hash function selected from the universal hash family. If we are in the non-adaptive setting, in which there is a fixed sequence of updates and a fixed point query about an element $x$ executed after all updates, then with probability $1 - \delta$, the returned estimate $\widehat{a_x}$ is at most $a_x + \epsilon \|\boldsymbol{a}\|_1$, where $\boldsymbol{a}$ is the frequency vector after all the updates.*

## 2.1 Hash Functions

The family of functions (or more generally, the distribution on functions) from which CMS selects each hash function plays a crucial role when it comes to attacking the estimates provided by CMS. Not all hash functions allow for breaking estimates for every element $x \in \Omega$. For example, if one of the hash functions puts $x$ in its own bucket (i.e., all other elements are mapped to other buckets), all estimates $\widehat{a_x}$ provided in reply to a point query about $x$ are exact, i.e., $\widehat{a_x} = a_x$, where $a_x$ is the frequency of $x$ at the query time.

We are mainly focusing on two models:

- The Random Oracle Model: This is an idealized scenario where each element gets a uniformly random hash value, independently of other assignments.

- Universal Hashing: This is a more realistic model where we put a constraint on how often the hashes of two elements can collide.

  A popular example of a family of universal hash functions mapping $[k]$ to $[B]$ is the family of functions $h(x) = ((a \cdot x + b) \bmod p) \bmod B$, spanning over all $a, b \in [p]$, where $p$ is a fixed prime such that $B \leq k \leq p$. Introduced by Carter and Wegman (1977), this family and its variations have has gained a lot of popularity in practice. It is the default for CMS in Apache Spark and Twitter Algbird.

Our techniques can also be applied, with relatively small modifications, to other models such as weakly uniform hash functions, generalizations of universal hashing and arbitrary hash functions. We explain these models, as well as more details about the two main models in Appendix B.

## 3 Making Count-Min Sketch Robust Against Adaptive Adversaries

In this section, we explore conditions under which Count-Min Sketch is adversarially robust. This allows for both making the data structure provide correct estimates with high probability and sets limits on what attacks can achieve.

We start by observing that if no two elements used by the user overlap in more than an $\epsilon$ fraction of the rows, then all estimates are $\epsilon$–estimates. In this case, no element can be attacked with fewer than $1/\epsilon$ elements and this, in particular, implies that the attack has to spread over several elements which can be used to limit the overestimate to at most $\epsilon \|\boldsymbol{a}\|_1$. Later we show conditions under which this kind of scenario is likely to occur, leading to only correct estimates being returned to the user.

**Lemma 3.** *Let $\epsilon \in (0, 1)$. Consider an interaction with a CMS instance in which all elements used by the user (in either updates or point queries) are such that for all pairs of them, the fraction of*

*rows in which they are mapped to the same bucket is less than $\epsilon$. Then all estimates produced in this interaction are $\epsilon$–estimates.*

*Proof.* If $\|\boldsymbol{a}\|_1 = 0$, then all elements have frequency 0, all buckets contain value 0, and the CMS instance trivially outputs the exact value. Hence we assume from now on that $\|\boldsymbol{a}\|_1$ is positive. Consider an element $x$ that is queried at some point. Let $\Delta_i$ be the overestimate in $x$'s bucket in row $i$ at the time of the query, where $i \in [R]$, i.e., the total weight of other elements assigned to this bucket at this specific moment. In order for $x$'s frequency to be overestimated by $\epsilon\|\boldsymbol{a}\|_1$ or more, its frequency has to be overestimated by at least $\epsilon\|\boldsymbol{a}\|_1$ in each bucket to which it is assigned, i.e., for all $i \in [R]$, $\Delta_i \geq \epsilon\|\boldsymbol{a}\|_1$. Then, by summing over all buckets, $\sum_{i=1}^{R} \Delta_i \geq \epsilon R\|\boldsymbol{a}\|_1$. We now argue that this cannot be true, i.e., overestimates by $\epsilon\|\boldsymbol{a}\|_1$ do not occur. Consider any $y \neq x$ that the algorithm uses and that has a non-zero frequency $a_y$ at the moment of query. Since it shares less than $\epsilon R$ buckets with $x$, it means that its contribution to the sum of overestimates $\Delta_i$ is less than $\epsilon R a_y$. By summing over all such elements $y$, we learn that the sum of all overestimates is strictly less than $\epsilon R\|\boldsymbol{a}\|_1$, which proves that the frequency of $x$ is not overestimated by $\epsilon\|\boldsymbol{a}\|_1$ or more. $\qquad\square$

## 3.1 THE RANDOM–ORACLE MODEL

We start by analyzing the performance of CMS in the random oracle model with a sufficient number of rows. In this model, the adversary cannot predict values of hash functions on elements she has not used yet, even if she knows values of hash functions on all elements she has already used while interacting with the data structure. Clearly she cannot learn more from the previous queries since this information uniquely determines answers to all queries.

We first state a specific version of the Chernoff bound that we use.

**Fact 4** (Chernoff bound). *Let $X_1$, $X_2$, …, $X_n$ be independent random variables with values in $\{0, 1\}$. Let $X = \sum_{i=1}^{n} X_i$ and let $\mu = E[X]$. For any $\epsilon \geq 1$, $\Pr(X \geq (1+\epsilon)\mu) \leq e^{-\epsilon\mu/3}$.*

We now show that no large overlaps between elements are likely to happen for a *fixed* set of elements as long as all hash functions are selected from a universal hash function family.

**Lemma 5.** *Let $\epsilon \in (0, 1]$, $\delta \in (0, 1)$, and $Q \in \mathbb{Z}_+$. Consider a CMS instance with $B \geq 2/\epsilon$ buckets per row and $R \geq \frac{12}{\epsilon}\ln Q + \frac{6}{\epsilon}\ln(1/\delta)$ rows. Suppose that the hash function in each row is selected independently from a universal hash function family. Consider a set $S$ of at most $Q$ elements. The probability that two of them are mapped to the same bucket in at least $\epsilon R$ rows is at most $\delta$.*

*Proof.* Consider two elements $x$ and $y$ in $S$. For each $i \in [R]$, let $X_i \in \{0, 1\}$ be the indicator variable whether $x$ and $y$ are mapped to the same bucket in row $i$. Due to the universality of hash functions, $\mathbb{E}[X_i] = B^{-1}$. Let $X = \sum_{i=1}^{R} X_i$. We have $\mathbb{E}[X] = R/B \leq \epsilon R/2$. The probability that $x$ and $y$ are in the same bucket in at least $\epsilon R$ rows can be bounded, using the Chernoff bound (see Fact 4): $\Pr[X \geq \epsilon R] \leq e^{-(\epsilon R - R/B)/3} \leq e^{-\epsilon R/6} \leq \delta/Q^2$. By the union bound, the probability that two elements in $S$ share more than $\epsilon R$ buckets is, therefore, bounded by $\binom{Q}{2} \cdot \frac{\delta}{Q^2} \leq \delta$. $\qquad\square$

We can now finally prove our main claim for fully random hash functions, i.e., the random oracle model.

**Theorem 6.** *Let $\epsilon \in (0, 1]$, $\delta \in (0, 1)$, and $Q \in \mathbb{Z}_+$. Consider a CMS instance with $B \geq 2/\epsilon$ buckets per row and $R \geq \frac{12}{\epsilon}\ln Q + \frac{6}{\epsilon}\ln(1/\delta)$ rows. If the number of times the CMS instance is accessed is bounded by $Q$ and the internal hash functions follow the random oracle model, then with probability at least $1 - \delta$, all estimates provided by the CMS instance are $\epsilon$–estimates.*

*Proof.* Every interaction by the adversary with the CMS instance, be it a query or update, involves some element $x$. We claim that the adversary does not learn more from all queries than the buckets to which each element the adversary used was assigned. This is the case, because knowing the bucket assignments uniquely determines the results of all queries, and frequency queries are the only way the adversary learns about the internal workings of the data structure. In the random oracle model, each bucket assignment is uniform and independent of the assignments of other element. This in particular means that the random oracle model has the universal hashing properties. Additionally, the specific labels the adversary uses do not matter in the random oracle model, since each element

is assigned independently of the assignment of other elements to buckets, so there is no gain to be achieved by selecting one new element versus another. If the adversary accesses the CMS instance $Q$ times, one can create a list of at most $Q$ unique elements, and whenever the adversary needs a new element, the next element on the list can be used. It, therefore, follows from Lemma 5 that with probability at least $1 - \delta$, no two of at most $Q$ elements that the adversary uses are assigned to the same bucket in $\epsilon R$ or more rows. And if this is the case, it follows from Lemma 3 that all estimates provided by the CMS instance are $\epsilon$–estimates. □

## 3.2 Universal Hashing

We now show that universal hashing is sufficient to ensure that no two elements collide on more than an $\epsilon$–fraction of rows if the size of the domain is at most polynomially greater than the number of buckets per row.

**Theorem 7.** *Let $k$ be a positive integer, $\epsilon \in (0, 1)$, and $\delta \in (0, 1/2]$. Let $S$ be the size of the domain of elements for which we want to provide approximate frequency counts. Consider a CMS instance with $R \geq 2\epsilon^{-1}k \log_S(1/2\delta)$ rows and $B \geq eS^{1/k}/\epsilon$ buckets per row in which all hash functions are independently selected from a universal hash function family. With probability at least $1 - \delta$ taken over the initial selection of hash functions, all estimates provided for point queries by CMS are $\epsilon$–estimates.*

*Proof.* Consider two different elements $x$ and $y$. Let us bound the probability that the fraction of rows in which they collide (i.e., share the same bucket) is at least $\epsilon$. This corresponds to the elements ending up in the same bucket in at least $t = \lceil \epsilon R \rceil$ rows. For any subset of $t$ rows, the probability that these two elements collide in all of them is bounded by $B^{-t}$ due to the universality and independence of hash functions. We now apply the union bound to bound the probability that this happens for some subset of $t$ rows. It is at most

$$\binom{R}{t}B^{-t} \leq \left(\frac{R \cdot e}{t}\right)^t \cdot B^{-t} \leq \left(\frac{e}{\epsilon B}\right)^{-t} \leq S^{-t/k} \leq S^{-\epsilon R/k} \leq S^{-2-\log_S(1/2\delta)} = \frac{2\delta}{S^2}.$$

Applying the union bound over all pairs of elements, we can bound the probability that this happens for some pair of elements by $\binom{S}{2}\frac{2\delta}{S^2} \leq \frac{S^2}{2} \cdot \frac{2\delta}{S^2} \leq \delta$. We now apply Lemma 3 to conclude that with probability $1 - \delta$ over the selection of hash functions, all estimates provided by the CMS instance are $\epsilon$–estimates. □

In particular, by plugging in $k = \lceil \log S \rceil$, we learn that this guarantee can be achieved as long as there are at least $2\epsilon^{-1}\lceil \log S \rceil \log_S(1/2\delta)$ rows and at least $2e/\epsilon$ buckets per row.

## 4 New attacks on CMS

In this section, we describe algorithms which, given access to a Count-Min Sketch *CMS* and an element *target*, return a set of elements that doesn't include *target* which, when inserted into the *CMS* will increase *target*'s count. Formally, these algorithms find a **cover set** of the target.

**Definition 4.1.** *A set of elements from the universe of possible values of the stream is called a **cover set** for* target *if it doesn't contain* target *and for each row $i$ of the CMS, it contains at least one element $x_i$ so that $h_i(x_i) = h_i(\text{target})$*

### 4.1 The General Approach and Algorithm

Unless stated otherwise, we assume the hashes follow the Random Oracle Model.

Given an element *target*, we can insert a stream of uniformly randomly selected elements $x_i$, and query *target* after every insertion until we get a returns a non-zero value. When this happens, we know that a subset of the inserted elements can increase *target*'s count, and that the last element inserted collides with *target* in at least one row.

One strategy we could employ is to re-insert this value that collided with *target* with a very large weight (called $\infty$ [3]), and then repeat inserting random elements one by one until the estimate increases

---

[3]In practice inserting $10^9$ suffices.

again, at which point we know we found an element that collides with *target* on a different row. We stop when the query returns a value $\geq \infty$, so when all rows had collisions.

This approach is similar to the one presented in Markelon et al. (2023) so we included the code (Algorithm 2) as well as observations about the expected runtime and cover set size in appendix D.

## 4.2 USING THE VALUES OF CMS ESTIMATES

**Definition 4.2** (Forcing row). *A row is called a **forcing row** for a given* target *if its value for* target *is minimal across all rows, and thus it matches the CMS estimate.*

Algorithm 2 does not take full advantage of the point query results or of being able to vary the weights of the updates. We propose a new algorithm that uses this information to build a cover set faster.

The new algorithm relies on the extra parameters $b$, which we will use as a base, and $y < x$, which we will use to increment values by uniformly random amounts $\in [b^x, b^x + b^y)$ instead of by 1. By keeping track of what element added what value, we can check if the estimate of the target is less than $2 \cdot b^x$. If it is, we know there is only one element in the forcing row (4.2), and we can recover that element. More generally if the value is less than $b^x \cdot c$ for $c < b^{x-y}$, there are exactly $c$ collisions in the forcing row.

**Claim 8.** *After inserting $k$ elements into a CMS, we need a total of $k$ bits to deterministically reconstruct which elements collided with the target.*

*Proof.* Every subset of elements has a non-zero probability to be part of the forcing row, so there are $2^k$ combinations, meaning that we need at least $k$ bits to uniquely recover the subset. $\square$

---

**Algorithm 1:** Value based CMS attack

**input** : *CMS*, an instance of a CMS; *target* a target element, $b, x, y$ as described above

1   *coverSet* $\leftarrow \varnothing$
2   *allElements* $\leftarrow \varnothing$
3   **while** *CMS.pointQuery*(*target*) *is* 0 **do**
4     $r \leftarrow$ *randomElement*()
5     $w \leftarrow$ *randomElement*($b^y$) $+ b^x$
6     *CMS.updateElementFrequency*($r, w$)
7     *allElements* $\leftarrow$ *allElements* $\cup \{(r, w)\}$
8     *lastElement* $\leftarrow r$
9   *CMS.updateElementFrequency*(*lastElement*, $\infty$) *coverSet* $\leftarrow$ *coverSet* $\cup \{$*lastElement*$\}$
10 **while** *canRecoverKey* (*CMS.pointQuery*(*target*), *allElements*) **do**
11     *recoveredElements* $\leftarrow$ *recoverElements*(*CMS.pointQuery*(*target*))
     *coverSet* $\leftarrow$ *coverSet* $\cup \{$*recoveredElements*$\}$
12     *CMS.updateElementFrequency*(*recoveredElements*, $\infty$)
13 **while** *CMS.pointQuery*(*target*) *is not* $\infty$           // Revert to Algorithm 2
14 **do**
15     *oldTargetValue* $\leftarrow$ *CMS.pointQuery*(*target*)
16     **while** *CMS.pointQuery*(*target*) *is oldTargetValue* **do**
17       *oldTargetValue* $\leftarrow$ *CMS.pointQuery*(*target*)
18       $r \leftarrow$ *randomElement*(*allElements*)
19       *CMS.updateElementFrequency*($r, 1$)
20     *coverSet* $\leftarrow$ *coverSet* $\cup \{r\}$
21 return *coverSet*

---

After the minimum value increases, if the increase is between $c \cdot b^x$ and $(c + 1) \cdot b^x$, we know that there are $c$ inserted elements that are forcing the minimum value. Let $n$ be the total number of elements inserted in this iteration. We can either look at all the $\binom{n}{c}$ possible combinations and find a subset of values adding up to this sum, or use a classical knapsack dynamic programming in $O(b^y \cdot c \cdot n + \text{SOL}_{\text{CNT}})$, where $\text{SOL}_{\text{CNT}}$ is the number of valid combinations, to retrieve all possible

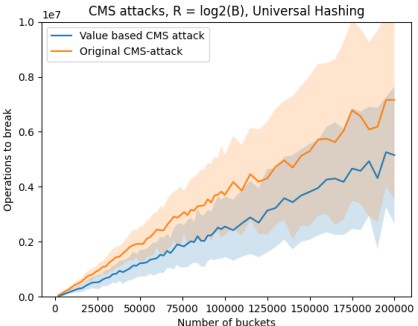 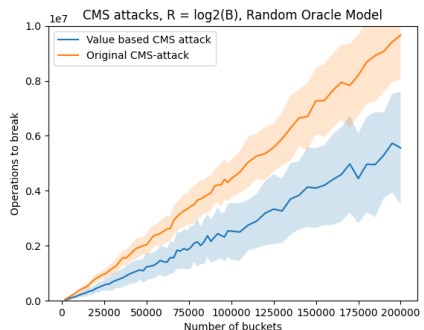

(a) Algorithm comparison, Universal Hash    (b) Algorithm Comparison, Random Oracle Model

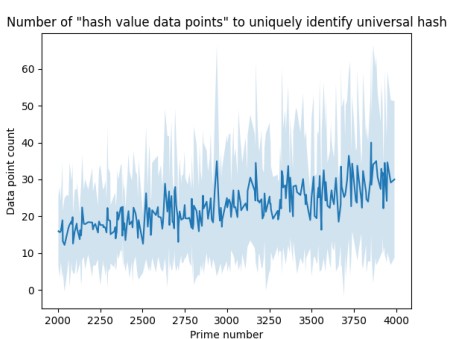 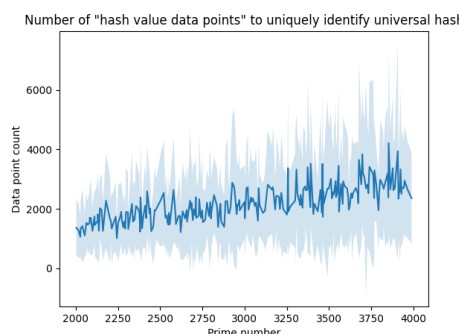

(a) Time to unique solution (data from Def. 5.1)   (b) Time to unique solution (data from Def. 5.2)

solutions. A final note is that these algorithms work on both Random Oracle Model, Universal hash functions and even Weakly uniform hash functions, since the only requirement for them to succeed is for the functions to have enough collisions with the target element.

# 5 EXPERIMENTAL RESULTS

## 5.1 CMS ATTACK EXPERIMENTS

We ran both Algorithm 2 and Algorithm 1 on a real black box CMS, where the attacker did not have access to the hash functions or size of the CMS. We set $B$ from 2000 to 200,000 in 2000 increments and 5000 after 100,000. For a fixed $B$ we have $R = \log_2(B)$, which is the usual practical setup used in the non-adversarial model to ensure a small error margin while still saving memory.

The attacks were run 100 times for each CMS bucket size, averaged, and plotted with the standard deviation shaded in the plots.

The plots show that in practice the value based attack uses from $20\%$ to $30\%$ fewer queries to find a cover set, and that universal hashing is slightly easier to attack than the random oracle model. We observed similar results when attacking the Apache Spark CMS implementation.

## 5.2 FINDING MULTI-HASH COLLISIONS

The only possible way to get a cover set smaller than $R$ is to have collisions in multiple CMS rows. The experiments in this section explore finding multi-hash collisions using CMS attacks.

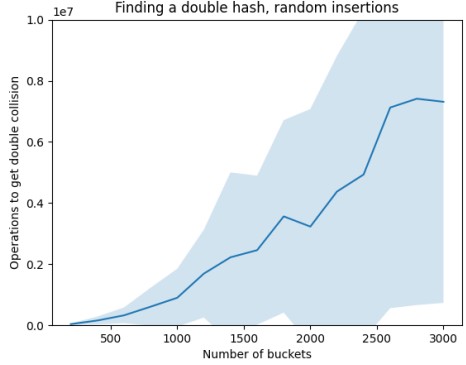
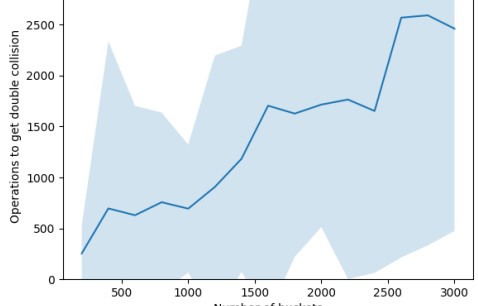

(a) Insertions needed with no information        (b) Insertions needed with a known hash

### 5.2.1 UNIVERSAL HASHING

In some CMS implementations using $((a \cdot x + b) \mod P)$ universal hash families, the prime $P$ is publicly available. We can also find $B$ and $R$ given black box access to a CMS using the methods we describe in appendix E.

It is possible to also find the values of $a$ and $b$ for one of the hashes using more attacks. This can be used to narrow down the number of attacks needed to find a value that collides with the target in both the known hash and another hash.

There are two types of information we can get about a hash function:

**Definition 5.1.** *Hash value data point: for one value we get $x$'s hashed value: $h(x)$.*

**Definition 5.2.** *Equality of hashes data point: for two values $x$ and $y$, we get whether $h(x) == h(y)$.*

The first type of data point is hard to obtain in practice, but is valuable as a bound for how long it would take to solve the problem in ideal conditions. The second type can be obtained from a black box CMS.

The experiment setup involved fixing the prime number and repeating the process of trying to find $a$ and $b$ by randomly sampling elements, getting either their hash value or whether they collide with a target, and using the observation to reduce the number of possible $(a, b)$.

The plots show the mean and the standard deviation for the number of samples needed to narrow down $a$ and $b$ across the runs plotted against prime numbers between $2000$ and $4000$. These values are smaller than what we would see in reality, but they still show that having access only to hash collisions requires a significant amount of observations, while having access to hash values can fully reveal the hash quickly.

Knowing that one of the hashes uses the formula $h(x) = ((a \cdot x + b) \mod P) \mod B$ allows us to enumerate all potential elements that collide with a given target: $(B + k \cdot h(\textit{target}) - b) \cdot a^{-1}, 0 \leq k \leq \frac{P}{B}$. Note that these elements are randomly distributed with respect to the other hashes, so the probability that they collide with the target in another row is $\frac{1}{B}$, meaning that in expectation we need to test $B$ of them until we find a second collision.

We ran two experiments to simulate this setup, using a CMS with 2 rows. In one setting, we insert elements randomly until reaching a double collision, while in the second one we simulate knowing the full formula for one of the hashes and inserting only elements where the known hash collides with with the target. The experimental results also show a reduction from $B^2$ (3a) to $B$ (3b) steps.

In appendix F we also discuss how an attacker can generate 2-hash collisions with full information about two functions, and we analyze the limitations of when 2-hash collisions exist. In particular, we prove that if the number of buckets is large enough, it is possible to have no 2-hash collisions. Note that in practice, the hashes would be sampled from a universal hash family, and not manually selected, so we should still expect 2-hash collisions to be possible.

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

## A  COUNT-MIN SKETCH ALGORITHM AND PROPERTIES

---

**Algorithm 1:** Count-Min Sketch$(R, B)$

---

**Variables:** array $A$ with $R$ rows and $B$ buckets per row

1  **routine** *initialization*:
2     **for** $r \in [R]$ **do**
3         $h_r \leftarrow$ randomly selected hash function from $\Omega$ to $[B]$
4         **for** $b \in [B]$ **do**
5            $A[r][b] \leftarrow 0$

6  **routine** *updateElementFrequency*$(x, \Delta)$:
7     **for** $r \in [R]$ **do**
8         $A[r][h_r(x)] \leftarrow A[r][h_r(x)] + \Delta$

9  **function** *pointQuery*$(x)$:
10     *estimate* $\leftarrow \infty$
11     **for** $r \in [R]$ **do**
12         *estimate* $\leftarrow \min\{estimate, A[r][h_r(x)]\}$
13     **return** *estimate*

---

**Proof of Fact 1:**    *The result of a point query in CMS is never an underestimate of the frequency of the queried item.*

*Proof.* Consider an arbitrary element $x$. The value stored in each bucket is the sum of non-negative frequencies of elements mapped to it. Hence for each bucket bucket $A[r][h_r(x)]$, $r \in R$, to which $x$ is mapped, the value stored in it is $a_x$, the frequency of $x$, plus the sum of non-negative frequencies of other items mapped to this bucket. Therefore, the value in each bucket to which $x$ is mapped is at least $a_x$, and since the result of the point query for $x$ is the minimum of values in buckets to which $x$ gets mapped, it has to be at least $a_x$ as well. $\qquad\square$

**Proof of Fact 2:**    *Let $\epsilon \in (0, 1)$ and $\delta \in (0, 1)$. Consider a CMS instance with $R \geq \lceil \frac{e}{\epsilon} \rceil$ rows and $B \geq \lceil \ln(1/\delta) \rceil$ buckets per row with each hash function selected from the universal hash family. If we are in the non-adaptive setting, in which there is a fixed sequence of updates and a fixed point query about an element $x$ executed after all updates, then with probability $1 - \delta$, the returned estimate $\widehat{a_x}$ is at most $a_x + \epsilon \|\boldsymbol{a}\|_1$, where $\boldsymbol{a}$ is the frequency vector after all the updates.*

*Proof.* The proof of this fact is relatively simple and follows in two steps. First, one proves that in each row, the probability that the weight of the bucket to which $x$ is assigned is more than $a_x + \epsilon \|\boldsymbol{a}\|_1$ is at most $1/e$ via Markov's inequality. To get a worse estimate overall, the estimates from all rows would have to be greater than $a_x + \epsilon \|\boldsymbol{a}\|_1$. Since the hash functions are selected independently for all rows, this cannot happen with probability greater than $e^{-R} \leq e^{-\ln(1/\delta)} = \delta$. $\qquad\square$

## B  HASHING MODELS

### B.1  THE RANDOM ORACLE MODEL

The random oracle model formalizes the idea of a fully random hash function. In other words, a hash function on this model is completely unpredictable. This is a very idealized scenario that is impossible to achieve in constant space, but it is used for analyzing cryptographically hard functions with a secret random seed, which behave essentially randomly for an adversary with bounded computational resources.

**Definition B.1.** *A hash function $h : \Omega \rightarrow Z$ in the* random oracle model*, where $Z$ is a finite set, assigns a uniformly random value from $Z$ as $h(x)$ for each $x$, independently of the assignment of other values.*

### B.2 UNIVERSAL HASHING

Universal hashing is a weaker (but also more realistic) model that can be implemented efficiently in practice and suffices for analyzing CMS in the non-adaptive setting (see Fact 2). In this model, we just put a constraint on how often the outputs of two elements can collide. This idea was introduced by Carter and Wegman (1977).

**Definition B.2.** *In* universal hashing*, a hash function* $h : \Omega \to Z$*, where $Z$ is a finite set, is selected uniformly from a family $\mathcal{F}$ of functions such that for any two different elements $x, y \in \Omega$,* $\Pr[h(x) = h(y)] \leq 1/|Z|$*. We refer to $\mathcal{F}$ with this property as the* universal hash function family*.

It is easy to see that a hash function in the random oracle model has the universal hashing property for $\mathcal{F}$ being simply the set of all functions from $\Omega$ to $Z$.

### B.3 OTHER MODELS

- **Weakly uniform hash functions:** Let $c > 0$ be a fixed constant. We say that a hash function $h : [k] \to [B]$ is *c–weakly uniform* if for all $x \in [k]$, the number of elements that are mapped to $h(x)$ (including $x$) is at least $ck/B$. If $ck/B \geq 2$, then it is possible to efficiently find a collision in every row for an element for which we want to make CMS provide a significant overestimate and it is possible to adjust our techniques to this scenario.

- **Generalizations of universal hashing:** A popular relaxation of universal hashing is to allow for the probability of collision of any two elements $x$ and $y$ to be at most $c/B$ for some $c \geq 1$. We later prove that universal hashing is sufficient for ensuring that a version of CMS is resistant against adaptive updates if $\Omega$ is not arbitrarily larger than the amount of available memory. This prove carries over to this model with slightly worse constants.

- **Arbitrary hash functions:** Finally, it is possible to apply our techniques to create a query for which CMS provides an overestimate by $1/R$ for an arbitrary set of hash functions. This may not work for making CMS overestimate the frequency of any given element (because it could be put in one of the buckets by itself), but a randomly selected element is likely to collide with many other elements in every row.

## C RELATIONSHIP TO PREVIOUS WORK ON ROBUSTNESS OF FREQUENCY ESTIMATORS

Clayton et al. (2019) consider some adversarial attacks on Count-Min Sketch, with the main focus being other types of randomized data structures such as Bloom filters and counting Bloom filters. For Count-Min Sketch, they make the observation that internal sketch cannot be know to the adversary. If it is known, the attacks become very easy to perform. To provide some amount of robustness, they introduce thresholdng, i.e., limit the number of non-zero entries in every row. Unfortunately, this could lead to significant underestimates, while in Count-Min Sketch frequencies can only be overestimated (see Fact 1). Additionally, Markelon et al. (2023), an intersecting set of authors, are very realistic about the shortcomings of this approach: "Further, a thresholding mechanism is used to achieve security for CMS, a solution that we deem untenable for real world uses of the CMS."

Markelon et al. (2023) focus on the insertion only setting. Their attack is based on finding a cover set by looking for collisions between different elements. Our attacks speed up the rate at which these kinds of collisions can be detect by recovering which elements collide from returned freqency estimates. We also show that multirow collisions (and therefore, worse estimates) can be achieved more efficiently for a popular class of universal hash functions. They also propose a data structure that is adversarially robust, but it can only handle insertions of elements. Indeed, the HeavyKeeper algorithm on which they build is similar to the Misra–Gries algorithm Misra and Gries (1982). It cannot be directly extended to handle deletions, because it relies on Misra–Gries–like forgetting of elements. Once an element is forgotten, it is hard to recover it and restore estimates with all theoretical guarantees for elements that were not deleted from the stream. Our analysis of Count-Min Sketch to make it robust to adversarial environments applies to streams of both insertions and deletions. Additionally, our approach does not require a new algorithm, other than adjusting parameters of Count-Min Sketch, the numbers of rows and buckets.

## D  COVER SET ALGORITHM WITHOUT USING QUERY VALUES

---

**Algorithm 2:** CMS attack

---

**input**  : *CMS*, *target* (a CMS instance and target element)
**output**: A set of elements that doesn't include *target*, that if we insert them into the CMS, this increases *target*'s value

1  *coverSet* ← ∅
2  *allElements* ← ∅
3  **while** *CMS.pointQuery(target) is* 0 **do**
4  |   *r ← randomElement()*
5  |   *CMS.updateElementFrequency(r, 1)*
6  |   *allElements ← allElements ∪ {r}*
7  |   *lastElement ← r*

8  *CMS.updateElementFrequency(lastElement, ∞) coverSet ← coverSet ∪ {lastElement}*
9  **while** *CMS.pointQuery(target) is not* ∞ **do**
10 |   *oldTargetValue ← CMS.pointQuery(target)*
11 |   **while** *CMS.pointQuery(target) is oldTargetValue* **do**
12 |   |   *oldTargetValue ← CMS.pointQuery(target)*
13 |   |   *r ← randomElement(allElements)*
14 |   |   *CMS.updateElementFrequency(r, ∞)*

15 |   *coverSet ← coverSet ∪ {r}*

16 return *coverSet*

---

**Claim 9.** *coverSet will have size $\leq R$.*

*Proof.* Every time we insert an element $r$ into the set, it was an element that collided with *target* in at least one row and has no occurrences inside *coverSet* (otherwise that row would be ∞). Afterwards by making its value be ∞, all of the rows where it collides with *target* will be covered. So we can map each element to a disjoint set of rows (the ones it made ∞), meaning that we have at most as many elements as we have rows, so at most $R$ elements.  □

**Lemma 10.** *In expectation, the number of elements inserted before target's value increases is $O(R \cdot B)$.*

*Proof.* For a single row, the probability of selecting the same bucket as the target is $\frac{1}{B}$. Thus the expected value of inserted elements is $B$. Since the *CMS* returns the minimum across all rows, we need to select the same bucket as the target in every row. Since the rows are independent, the expected number of total inserted elements is $O(B \cdot R)$.  □

## E  RETRIEVING INTERNAL CMS SIZE

As mentioned previously a standard CMS implementation has an internal 2d array of size $R \times B$, and figuring out what $R$ and $B$ are can expose other avenues of exploiting the hash.

**Definition E.1.** *A Wide CMS is a CMS where the number of buckets ($B$) is significantly larger than the number of rows, so $B >>> R$.*

In this section we assume we're targeting a Wide CMS, which is a natural assumption since this is how they're mostly used in practice. Given standard access, we'll try to recover both $R$ and $B$.

### E.1  RECOVERING $B$

To compute the size of $B$, we can get a *coverSet* from Algorithm 1, then insert all the elements from the list except the last one. Afterwards, we insert random elements until *target*'s value changes. Since with high probability the element that we ignored was only matching *target's* hash inside one row and in the random oracle model every bucket has probability $\frac{1}{B}$, in expectation we'll need $B$ elements inserted until the *target*'s value changes. We can repeat this process and output the average.

**Fact 11** (Chernoff bounds). *Let $X_1, X_2, \ldots, X_n$ be independent random variables with values in $\{0,1\}$. Let $X = \sum_{i=1}^{n} X_i$ and let $\mu = E[X]$. Then for any $\delta \in (0,1)$:*

$$Pr[X > B \cdot T(1+\delta)] \leq \left( \frac{e^{\delta}}{(1+\delta)^{1+\delta}} \right)^{\mu}$$

$$Pr[X < B \cdot T(1-\delta)] \leq \left( \frac{e^{-\delta}}{(1-\delta)^{1-\delta}} \right)^{\mu}$$

Let's now show that the probability to be within $\delta$ of the correct number of buckets is sufficiently large. Let's assume we use $T$ iterations and get estimates $X_1, .., X_T$ and define $X = \sum_{i=1}^{T} X_i$. We know that $E[X] = B \cdot T$, so we can use this version of Chernoff bounds (Fact 11) with $\mu = B \cdot T$

In practice, if we want our error margin $\delta = 0.01$ with probability at least $99\%$, we could pick T in such a way that the two bounds are both smaller than $0.005$.

$$\frac{e^{0.01}}{1.01^{1.01}} \approx \frac{e^{-0.01}}{0.99^{0.99}} \approx 0.99995 \implies B \cdot T \geq \log 0.005 / \log 0.99995 \approx 106,000$$

---

**Algorithm 3:** Infer size of $B$

---

1   *CMS*, *target*, Instance of Count-Min Sketch, target element
2   *coverSet*, A set of elements that increase target's count.
3   *estimate* $\leftarrow 0$
4   $T \leftarrow 100.000/|coverSet| * const$
5   **for** $u \in [T]$ **do**
6      clear(*CMS*)
7      shuffle(*coverSet*)
8      $CMS.insert(coverSet[0...size-2])$
9      **while** $CMS.pointQuery$ (*target*) *is* 0 **do**
10         $CMS.insert(randomElement())$
11         *estimate* $\leftarrow$ *estimate* $+ 1$

12   return $\frac{estimate}{T}$

---

### E.2   Recovering $R$

Now let's think about inferring $R$ assuming that we know $B$.

**Definition E.2.** *Let $E(B, R)$ be equal to the expected number of elements we need to add to a CMS of size $B \times R$ so that target's value increases 1.*

We will use a similar approach to the previous algorithm: we'll start with an empty CMS and insert elements until *target*'s value is 1. We'll do this a number of times, take the average, and choose the $R$ so that $E(B, R)$ is the closest one to our average. Now we'll show how to compute $E(B, R)$.

**Definition E.3.** *Let $E_{B \times R}[i]$ be the expected number of elements we have to add in a $R \times B$ CMS, if there are still $i$ rows that do not have a hash collision with target.*

Since the initial CMS is empty and we need collisions in every row, the value we are looking for $E_{B \times R}[R]$ is describing exactly $E(B, R)$. We can write a system of $R$ equations with $R$ variables, then use Gaussian elimination to compute $E_{B \times R}[R]$.

$$E_{B \times R}[x] = 1 + \sum_{i=0}^{i=x} (1 - \frac{1}{B})^i * \frac{1}{B}^{x-i} * \binom{x}{i} * E_{B \times R}[i]$$

Applying naive Gaussian elimination on this will take $O(R^3)$, but we can notice that the equation dependencies form a directed-acyclic graph, so it can be computed in $O(n + m)$ where n is the number of states and m is the number of possible transitions, so in our case it will result in $O(R^2)$.

An interesting implementation detail is the fact that computing the combinations will not work, since their values will be too big to store in 64 bit integers. But the formula overall doesn't become too big, because the fractions that are multiplied with each combination, will overpower it and overall they're going to be small. We can rewrite the formula in the following way:

$$f(i, x) = \left(1 - \frac{1}{B}\right)^i * \frac{1}{B}^{x-i} * \binom{x}{i}$$

$$\implies f(i, x) = (1 - \frac{1}{B}) * f(i - 1, x) + \frac{1}{B} * f(i - 1, x - 1)$$

The base cases are $f(i, 0) = (1 - \frac{1}{B})^i$ and $f(i, i) = \frac{1}{B}^i$

Our final formula no longer depends on combinations, and it will not overflow

$$E_{B \times R}[x] = 1 + \sum_{i=0}^{i=x} f(i, x - i) * EX[i]$$

Now if we get a good estimate for $B$, our estimate will also be close for $R$, by following the same process of simulating multiple times how long it takes to increase the target element's value by 1 and averaging the results. Similarly to $B$, we would use Chernoff bounds (11 to pick the number of runs in order to satisfy a target approximation factor with high probability.

# F    MULTI-HASH COLLISIONS

## F.1    2-HASH COLLISIONS IN UNIVERSAL HASH FAMILIES

Let's analyze what happens if the attacker somehow gets full access to the hash functions. In particular, we want to answer if it is possible to find attack values that collide with the target value for multiple hash functions.

For simplicity, we look at 2 functions from a universal hash family:

$$h_1 = ((a_1 \cdot x + b_1) \bmod P) \bmod B$$

$$h_2 = ((a_2 \cdot x + b_2) \bmod P) \bmod B$$

Note that if $P > B^2$, we will have at least a pair of keys that collide on both hashes due to the pigeonhole principle. However, this doesn't necessarily guarantee that we can find a key that collides with our specific target, but the smaller the $B$ the more likely it is that we can find double collisions, as shown in figure 4b

For $P < B^2$ we make the following claim:

**Claim 12.** *If $P < B^2$, there exists a pair of hashes so that so that no two potential keys between 0 and $P - 1$ collide for both hashes.*

**Proof of Claim 12:**    *If $P < B^2$, there exists a pair of hashes so that so that no two potential keys between 0 and $P - 1$ collide for both hashes.*

*Proof.* Two such hashes can be obtained by setting $a_1 = 1$, $a_2 = \lceil \frac{P}{B} \rceil$ and $b_1 = b_2 = 0$.

Let's assume that we can find $0 \leq x < y < P$ so that:

$$(x \bmod P) \bmod B = (y \bmod P) \bmod B \iff x \bmod B = y \bmod B \iff y = x + k \cdot B$$

$$\left(\lceil \frac{P}{B} \rceil \cdot x \bmod P\right) \bmod B = \left(\lceil \frac{P}{B} \rceil \cdot y \bmod P\right) \bmod B$$

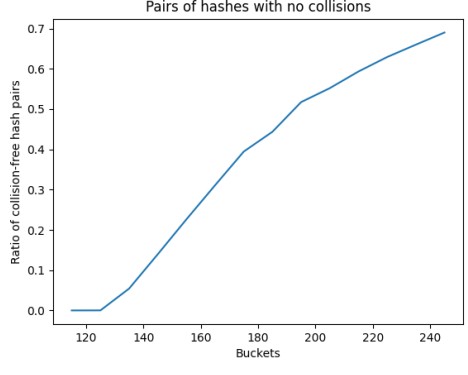

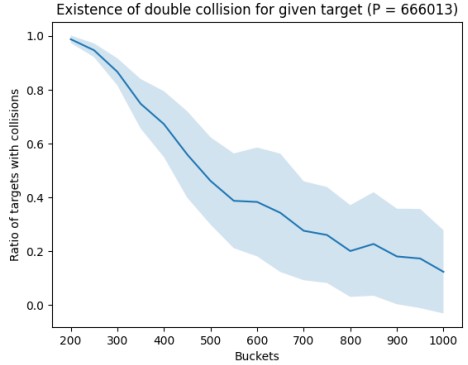

| (a) Collision-free pairs (P = 15377) | (b) Double collision for given target (P = 666013) |

Let:

$$\lceil \frac{P}{B} \rceil \cdot x = c_x \cdot P + r_x$$

$$\lceil \frac{P}{B} \rceil \cdot (x + k \cdot B) = c_y \cdot P + r_y$$

$$\implies \lceil \frac{P}{B} \rceil \cdot k \cdot B = P \cdot (c_y - c_x) + (r_y - r_x)$$

Note that $B$ divides $r_x - r_y$, so it must also divide $(c_y - c_x)$.

Since $x, y < P$, we have $k < \frac{P}{B}$. We use this and the fact that $P < B^2$ to obtain:

$$\lceil \frac{P}{B} \rceil \cdot k \cdot B < \lceil \frac{P}{B} \rceil \cdot \lfloor \frac{P}{B} \rfloor \cdot B \le P \cdot B$$

$$\implies P \cdot (c_y - c_x) < P \cdot B \implies c_y - c_x < B \implies c_x = c_y$$

$$\implies \lceil \frac{P}{B} \rceil \cdot k \cdot B = (r_y - r_x) \implies (r_y - r_x) > \lceil \frac{P}{B} \rceil \cdot B > P$$

This is not possible since they are both remainders of dividing numbers by $P$, so their difference is at most $P - 1$. Since we reached a contradiction by assuming $x$ and $y$ exists, this means that the 2 functions we chose always produce different pairs of hashes, which completes our proof. $\qquad\square$

We described our proof for one particular pair of hashes, but note that in most scenarios, the values of $a_1$, $b_1$ and $b_2$ can be chose arbitrarily as long as $a_2 = \lceil \frac{P}{B} \rceil \cdot a_1$. Additionally, as $B$ increases, there will be even more candidates for $a_2$ that result in collision-free hash pairs, as seen in figure 4a.

### F.2 2-HASH COLLISIONS WITH DIFFERENT PRIME NUMBERS

We also explored if it's possible to find 2-hash collisions if the functions take the form

$$((a_1 \cdot x + b_1) \bmod P_1) \bmod B$$

$$((a_2 \cdot x + b_2) \bmod P_2) \bmod B$$

There are $\frac{P_1}{B}$ possible values for $x$ modulo $P_1$ and $\frac{P_2}{B}$ possible values for $x$ modulo $P_2$ and each pair can generate a number between $0$ and $P_1 \cdot P_2$ using the Chinese Remainder Theorem. The downside is that we don't have enough information about the distribution of these solutions, and it is possible that all of them are outside the bounds of possible keys - recall that $P_1$ and $P_2$ are both larger than the universe of possible keys.

## F.3  RANDOM ORACLE MODEL

In the case of $R = 2$, an attack element would need to have a collision with the target for both hashes. Since the hashes are independent and each bucket has a $\frac{1}{B}$ probability of being chosen, the probability of a 2-hash collision is $\frac{1}{B^2}$. Thus, the expected number of random samples needed to find a value that increases both hashes (which is a cover set for $R = 2$) is $B^2$.

For the general case, we can compute probability of an element to collide for at least 2 of the $R$ hashes by subtracting the probability of no collisions and of exactly one collision from the total:

$$f(R) = \frac{B^R - (B-1)^R - R*(B-1)^{R-1}}{B^R} \approx \frac{\binom{R}{2}}{B^2}.$$

We could construct a cover set of size $\frac{R}{2}$ using the following algorithm: sample elements until finding a 2-hash collision, then repeat the process with the $R - 2$ remaining hashes. For simplicity let's assume $R$ is even. This gives us the following upper bound for the expected number of steps:

$$\frac{1}{f(R)} + \frac{1}{f(R-2)} + \frac{1}{f(R-4)} + \cdots \frac{1}{f(2)} \approx B^2 \sum_{i=1}^{\frac{R}{2}} \frac{1}{\binom{2i}{2}} < 2 \cdot B^2$$

