# OpenReview forum: "Adversarial Robustness of Count-Min Sketch"
_ICLR.cc/2025/Conference — Submitted to ICLR 2025_

### Official Review · Reviewer_RCp7 · 2024-10-27

**Soundness:** 1
**Presentation:** 1
**Contribution:** 2
**Rating:** 3
**Confidence:** 4

**Summary:**

This paper studies adversarial robustness for the CountMin data structure. Whereas roughly $(\log Q)/\varepsilon$ space is generally needed to answer $Q$ frequency estimation queries to an additive $\varepsilon\cdot L_1$ error in either the standard setting or the adversarially robust insertion-only setting, the best known adaptation for the adversarially-robust strict turnstile setting requires roughly $\sqrt{Q}/\varepsilon$ buckets. This paper shows that roughly $(\log Q)/\varepsilon^2$ space suffices. The paper also introduces an attack on CountMin, different than currently existing attacks. Finally, the paper provides experimental details to support the main theoretical results.

**Strengths:**

+ Adversarial robustness is an increasingly relevant model and frequency estimation is an important problem
+ This paper introduces a simple and elegant attack on CountMin with a relatively small number of queries
+ Robustness is claimed using space that only depends logarithmically in the number $Q$ of queries, compared to naively using existing results, which would give $\sqrt{Q}$ dependency

**Weaknesses:**

- Similar results are known for the insertion-only model and the robustness results in this paper do not apply to vectors with negative entries, so the regime of improvement is only for the strict turnstile model
- Attacks of similar (theoretical) quality are known from existing work by Markelon et. al. (2023)

**Questions:**

In the proof of Theorem 6, the claim is that the adversary does not learn more from all queries than the buckets to which each queried element was assigned. I'm not sure I see why this is true. In particular, why can the adversary not also learn that certain non-queried elements are NOT assigned to those buckets as well?

---

### Official Review · Reviewer_DP7c · 2024-11-02

**Soundness:** 2
**Presentation:** 2
**Contribution:** 2
**Rating:** 3
**Confidence:** 3

**Summary:**

The paper examines the robustness of the Count-Min sketch, a widely used data structure for frequency estimation, in the face of adaptive and potentially adversarial inputs. It considers two scenarios: one in which the internal hash functions employed by the Count-Min sketch are fully random, and another where the hash functions are universal. The study not only introduces methods for enhancing the resilience of the Count-Min sketch against adversarial inputs but also presents strategies for effectively attacking it under both models. Notably, both the defense and attack methods proposed are more efficient than those in previous works.

**Strengths:**

1. The problem addressed in this paper is both timely and significant, focusing on the resilience of data structures against adversarial scenarios.
2. The proposed data structure in this paper demonstrates a considerable improvement in space efficiency compared to prior adversarial-robust data structures.

**Weaknesses:**

1. I have some reservations regarding certain mathematical arguments in the paper:

   *  In Theorem 6, under the fully random hash function assumption, the paper claims: “If the adversary accesses the CMS instance $Q$ times, one can create a list of at most $Q$ unique elements, and whenever the adversary needs a new element, the next element on the list can be used.” Given this, why can't we rely on the original analysis of the Count-Min sketch, requiring $\frac{2}{\epsilon}$ buckets per row and $\ln\left(\frac{Q}{\delta}\right)$ rows to guarantee accurate frequency estimation for $Q$ elements with a failure probability at most $\delta$? Note that the original analysis of the Count-Min sketch provides stronger guarantees compared to the one presented in this paper. It would be beneficial for the authors to explicitly compare their analysis to the original Count-Min sketch guarantees and clarify why a different approach is required in adversarial settings. Doing so would highlight the novelty and necessity of their contributions more effectively.



   * In Theorem 7, is the set $S$ adaptively chosen by the adversary or predetermined in advance? Could you explicitly state whether $S$ is adaptively chosen or predetermined? This distinction is crucial as it significantly impacts the interpretation and implications of the theorem.


2. The presentation in the paper appears incomplete. For instance, in Algorithm 1, the meanings of “canRecoverKey” and “oldTargetValue” are not defined or explained, which could lead to confusion for readers. It is recommended to include definitions or explanations for these terms within the algorithm description or in a separate glossary.

**Questions:**

See weakness.

---

### Official Review · Reviewer_1HMw · 2024-11-02

**Soundness:** 3
**Presentation:** 3
**Contribution:** 3
**Rating:** 8
**Confidence:** 4

**Summary:**

This paper studies the CountMin sketch, a popular sketch for heavy hitters/frequency estimation, in the adaptive setting. Study of adversarial/adaptive sketching and streaming algorithms has received recent interest starting with the work of Hardt and Woodruff in 2013 and continuing with several papers in the past four years. The specific setting of this paper is that there is a fixed CountMin sketch in the strict turnstile model (insertions and deletions, but no negative frequencies). An adaptive adversary gives a sequence of updates and frequency queries to the sketch, receiving as feedback the standard CountMin estimate to its queries. The authors show positive and negative results in this model. Observe that, if there are no deletions, the determinitic Misra-Gries algorithm solves the same problem and is automatically adversarially robust as it uses no randomness.

Let $n$ be the size of the universe of stream elements. For positive results, the authors show that, if the hash functions are fully random, then setting the number of rows of the sketch to be $O(\log (n/\delta)/\epsilon)$ and the width of each row to be $O(1/\epsilon)$ guarantees that, with probability $1-\delta$, there does not exist any input for which any of the CountMin frequency estimates differ from the true frequency by more than $\epsilon$ times the $\ell_1$ norm of the stream. The authors present this result slightly differently, but this is a direct result of their proof and is the more interesting version, in my opinion. This result shows that adversarial robustness can be achieved by blowing up the standard number of rows by a $1/\epsilon$ factor. They also show that if the hash functions are only universal, then the same result can be achieved if the width of each row scales polynomially with $n$.

Both results follow from a nice observation that, in order to mess up the estimate of a CountMin sketch, there needs to exist a pair of keys that map to the same bucket in more than an $\epsilon$ fraction of the rows. If not, then increasing the estimate at any specific key by 1 requires increasing the norm of the stream by $\epsilon$. Then, basic probabilistic analysis gives the result by bounding how many rows are needed for this property to hold for all pairs of keys from a universe of size $n$.

The authors also describe attacks against CountMin sketches which utilize few rows (this is common in some applications) and show experimental results in attacking such sketches.

**Strengths:**

The positive results in this paper are simple and surprising. On the one hand, the Misra-Gries algorithm demonstrates that adversarial robustness is possible for $\ell_1$ heavy hitters (at least in the insertion-only case). On the other hand, many recent works on adversarial robust sketching/streaming shows that a polynomial dependence on the number of adversarial queries or on the universe size is required in order to achieve robustness. By contrast, in the random oracle model, the authors show that the only blowup needed is in the error parameter of the sketch. The argument is clean and can be extended to various adversarial settings as they show that with good probability, there does not exist any input data that could mess up the sketch.

**Weaknesses:**

The results in this paper are very straightforward, but they are interesting, especially in context of the difficulty (formally, via lower bounds) in making related sketches robust. To that end, I think that a more detailed discussion comparing and contrasting to related work on adversarially robust sketching/streaming is missing.

## Connections to Related Work

The adversarial model studied in this paper is not standard in the adversarial sketching/streaming literature. To my knowledge, there are two main settings which are studied in prior work.
1) Adversarial robust sketching: As in Hardt and Woodruff (2013) or the adversarial robustness of CountSketch in Cohen et al. (2022, 2023) is that a fixed sketch is queried with a vector (corresponding to an entire stream) and then an estimate is returned to the adversary (an estimate of a norm of the vector or a list of heavy hitters, respectively).
2) Adversarial robust streaming: As in Ben-Eliezer et al. (2022), a streaming algorithm continuously reports an estimate (e.g., of the number of distinct elements) while the adversary chooses the next stream elements.

The results of this work can extend to either case due to the nice observation that, with enough rows, with high probability, there does not exist any input that can corrupt the frequency estimation property of the sketch. I think the paper would be strengthened by making this stronger result more clear. A more concrete connection to these other works as well as a detailed discussion as to why CountMin can be made robust much more easily than, say CountSketch, would improve the paper. For the latter point, the fact that CountMin cannot be used for norm estimation, normally a downside, seems related to the fact that it can more easily be made robust.

## Small comments

Under "Our results: securing CMS", there is a typo in the first paragraph where $1/\epsilon^{-1}$ should be $1/\epsilon$. There is a typo in the second paragraph where a $1/\epsilon$ is replaced with $\epsilon$.

$\delta$ is overused in Section 2 as the increment amount in addition to the failure probability.

At the top of page 6, there is a typo, the exponent of $B$ should be $-t$ not $-k$.

**Questions:**

None

---

### Official Review · Reviewer_kanx · 2024-11-04

**Soundness:** 2
**Presentation:** 2
**Contribution:** 2
**Rating:** 3
**Confidence:** 4

**Summary:**

This work investigates the robustness of the Count-Min Sketch (CMS), a hash table-based data structure designed for memory and time-efficient frequency estimation, under adversarial attacks, where query sequences can be adversarially chosen rather than fixed.

The paper studies two settings with two models of hashing functions used to implement CMS: the random oracle model and universal hashing. The paper shows the number of rows and the number of buckets per row the hash table needs in both settings to make CMS adversarially robust, while ensuring an additive $\epsilon \|\|\textbf{a}\|\|_1$ accuracy of the query, where $\textbf{a}$ is the vector corresponding to the frequency of all items at the query time. Notably, in the more practical setting of universal hashing, the size of the domain (of items to be hashed) cannot be too large (polynomial) compared to the size of the buckets per row in the hash table of CMS.

On the attack side, this work considers one specific type of attacks where one inserts a set of items excluding a chosen target in CMS, while increasing the count of the chosen target in CMS. This work slightly extends the attack approach in [Markelon et al. (2023)] and proposes to smartly choose the items to be injected in CMS to make the attack faster.

Finally, this work empirically compares their attack approach against that in [Markelon et al. (2023)] to demonstrate the efficiency of the proposed approach, and further explores the efficiency of attacks under varying degrees of knowledge available to the attacker about the underlying hash functions.

**Strengths:**

CMS is a widely used data structure for fast frequency estimation. Exploring adversarial attacks on CMS is of practical significance.

**Weaknesses:**

$\textbf{1. The paper’s presentation is confusing at several places. }$

Some notations, without proper introduction, make it hard to follow the text. For instance, the frequency vector $\textbf{a}$ appears in the second paragraph of “Our results: securing CMS” without prior introduction, only being explained in the subsequent paragraph. Additionally, in Section 4.2, where the attack is described, it is unclear what “$Random(b^y)$” or “$n$” represent. The plots in the experiments in section 5 are not labeled properly, adding to the confusion.

At the beginning of Section 4.1, the paper states, “we assume the hashes follow the Random Oracle Model”; however, by the end of Section 4.2, it mentions that “these algorithms work on both the Random Oracle Model, Universal hash functions, and even Weakly uniform hash functions.” This inconsistency creates ambiguity regarding which hash function models the proposed attack applies to in Section 4.

The problem settings are not clearly or formally stated. On the robustness side, the notion of adversarial robustness in Section 3 is not formally defined. Similarly, on the attack side, key elements such as the attacker’s capabilities, goals, and necessary assumptions are not formally outlined. This lack of problem definition makes it hard to understanding the results.

The experiment section is also hard to follow. The main messages are not clear. The result plots are not properly referred to in the text. Section 5.2.2 seems a bit detached to the rest of the section.

$\textbf{2. Limited novelty. }$

The study of adversarial robustness of CMS in section 3 seems to be a straightforward application of existing techniques. The attack proposed in section 4 builds incrementally on the approach by [Markelon et al. (2023)]. There does not seem to be novel defense / attack methods, or new applications of existing techniques.

$\textbf{3. Comparison to prior work is not clear. }$

The paper references several prior works on making CMS adversarially robust, notably [Hassidim et al. (2022)], which introduces differential privacy to protect the internal randomness of CMS. However, the accuracy guarantees provided in [Hassidim et al. (2022)] differ from those in this paper. Specifically, [Hassidim et al. (2022)] achieves a stronger $(1\pm\alpha)$ multiplicative accuracy with probability $1-\delta$, whereas this paper targets a weaker additive accuracy guarantee of $\alpha ||\textbf{a}||_1$, where $\textbf{a}$ represents the frequency vector at the time of the query. This discrepancy makes the comparison of the CMS size, as discussed in the first paragraph of page 2, unfair.

Moreover, the paper does not clearly explain how the number of rows and buckets (i.e., the size of the CMS) in its results generally compares to that of prior work. In the related work section, the paper mentions that previously proposed robust CMS designs typically do not support deletion operations. If this is the reason that makes the results non-comparable, it would greatly benefit readers if the paper explicitly explained why deletion poses such a challenge and how this issue is addressed in the current study. Even with this caveat, it would still be valuable to outline how the size of a robust CMS differs when supporting deletions compared to prior work.

On the attack side, the paper asserts that its proposed modification to the approach by [Markelon et al. (2023)] is more efficient. However, it is unclear how the runtime of this modification theoretically compares to that of [Markelon et al. (2023)]. Or, is this improvement purely an empirical observation? This is confusing in the introduction. If the improvement is purely empirical (which seems so based on the rest of the paper), the main results regarding the percentage of runtime improvement (across different hashing functions) should be clearly summarized to provide a comprehensive understanding of the contribution.

**Questions:**

The main questions I have are:

1. How does the size of the robust CMS (i.e., the number of rows and buckets) described in Section 3 compare to that of previously proposed robust CMS, assuming the same accuracy guarantee?

2. How does the runtime of the proposed modification to the attack compare with the prior work by [Markelon et al. (2023)]?

3. In the experiments, several plots indicate that launching successful attacks requires applying on the order of $10^7$ operations to CMS. How practical are such attacks in real-world scenarios?

---

### Meta-Review · Area_Chair_Vgus · 2024-12-20

**Metareview:**

The paper considers the count-min sketch algorithm and examines whether its estimates are robust to an adversarial sequence. The problem is at an interesting intersection of questions around streaming algorithms in TCS and issues around adversarial robustness in ML. The analysis seems intuitive and interesting. However, many of the reviewers raised concerns and questions about relation to prior work, which the authors did not respond to. I suggest that the authors incorporate questions raised by the reviewers in their next version, and also the improvements suggested by Reviewer 1HMw. I suspect the paper will be above bar with these changes, but I do not recommend it being accepted at present.

**Additional Comments On Reviewer Discussion:**

There was no rebuttal from the authors.

---

### Decision · Program_Chairs · 2025-01-22

Reject